# Retention of Human iPSC-Derived or Primary Cells Following Xenotransplantation into Rat Immune-Privileged Sites

**DOI:** 10.3390/bioengineering10091049

**Published:** 2023-09-06

**Authors:** Thomas Später, Giselle Kaneda, Melissa Chavez, Julia Sheyn, Jacob Wechsler, Victoria Yu, Patricia Del Rio, Dave Huang, Melodie Metzger, Wafa Tawackoli, Dmitriy Sheyn

**Affiliations:** 1Orthopaedic Stem Cell Research Laboratory, Cedars-Sinai Medical Center, Los Angeles, CA 90048, USA; thomas.spaeter@cshs.org (T.S.); giselle.kaneda@cshs.org (G.K.); melissa.chavez2@cshs.org (M.C.); julia.sheyn@cshs.org (J.S.); jacob.wechsler@cshs.org (J.W.); victoria.yu@cshs.org (V.Y.); patricia.delrio@cshs.org (P.D.R.); wafa.tawackoli@csmc.edu (W.T.); 2Board of Governors Regenerative Medicine Institute, Cedars-Sinai Medical Center, Los Angeles, CA 90048, USA; 3Orthopedics Biomechanics Laboratory, Cedars-Sinai Medical Center, Los Angeles, CA 90048, USA; dave.huang@cshs.org (D.H.); melodie.metzger@cshs.org (M.M.); 4Department of Orthopedics, Cedars-Sinai Medical Center, Los Angeles, CA 90048, USA; 5Department of Surgery, Cedars-Sinai Medical Center, Los Angeles, CA 90048, USA; 6Department of Biomedical Sciences, Cedars-Sinai Medical Center, Los Angeles, CA 90048, USA

**Keywords:** xenotransplantation, intervertebral discs, mesenchymal stem cells, nucleus pulposus cells, inflammation, cell therapy, immune privilege, joint, PTOA

## Abstract

In regenerative medicine, experimental animal models are commonly used to study potential effects of human cells as therapeutic candidates. Although some studies describe certain cells, such as mesenchymal stromal cells (MSC) or human primary cells, as hypoimmunogenic and therefore unable to trigger strong inflammatory host responses, other studies report antibody formation and immune rejection following xenotransplantation. Accordingly, the goal of our study was to test the cellular retention and survival of human-induced pluripotent stem cell (iPSCs)-derived MSCs (iMSCs) and primary nucleus pulposus cells (NPCs) following their xenotransplantation into immune-privileged knee joints (14 days) and intervertebral discs (IVD; 7 days) of immunocompromised Nude and immunocompetent Sprague Dawley (SD) rats. At the end of both experiments, we could demonstrate that both rat types revealed comparably low levels of systemic IL-6 and IgM inflammation markers, as assessed via ELISA. Furthermore, the number of recovered cells was with no significant difference between both rat types. Conclusively, our results show that xenogeneic injection of human iMSC and NPC into immunoprivileged knee and IVD sites did not lead to an elevated inflammatory response in immunocompetent rats when compared to immunocompromised rats. Hence, immunocompetent rats represent suitable animals for xenotransplantation studies targeting immunoprivileged sites.

## 1. Introduction

Throughout the past decades, human mesenchymal stromal cells (MSCs) and other primary cells have emerged as a source for the development of novel cell therapies [1,2,3,4,5,6]. MSCs are considered hypoimmunogenic and therefore do not trigger a strong inflammatory response [7], making them an ideal candidate for allogeneic transplantation. To test these new therapies in animal models, previous studies have shown that the xenotransplantation of human MSCs in rat models promotes calvarial bone regeneration [8,9], an enhanced recovery from spinal cord injuries [10] and significantly improve the restoration of cardiac functions in pigs following myocardial infarction [11].

Other studies reported severe immune responses following xenogeneic transplantation of human MSCs into rats suffering from critical size femoral defects [12] and calvarial bone defects [8]. Such differing results indicate that the capability of human cells to function across species barriers may depend on various factors, such as the transplantation site and vascularization.

The immune system varies significantly between different tissues and organs of the same organism [13,14]. Vulnerable regions of the body, such as the central nervous system, the eyes and the brain, are concealed from the immune system to protect them from damage by overwhelming inflammatory responses directed against pathogens [15,16] and are thus considered to be immune privileged. However, the immune privilege status of different organs is not always clear. For example, cartilage tissue is considered an immune privileged site [17] although previous research demonstrated xenogeneic chondrocytes were strongly rejected by the host [18]. These varying results indicate that more research on the immune privilege status of certain tissue types is needed to clarify whether they will be rejected following an allogeneic or xenogeneic implantation of relevant target cells.

To investigate the cellular mechanisms related to post-traumatic osteoarthritis (PTOA), non-invasive knee injury is commonly used to rupture the anterior cruciate ligament (ACL) [19,20]. This knee destabilization not only serves as an indispensable prerequisite for proper development of PTOA but also activates the immune system.

In the current study, we investigated the retention and survival of human-induced pluripotent stem-cell-derived MSCs (iMSCs) and primary intervertebral discs (IVD)-derived nucleus pulposus cells (NPCs) injected into the knee joints following insult to the joint by tearing the anterior cruciate ligament (ACL) and IVDs of rats, respectively. NPCs have widely been investigated and described as promising cells to regenerate the IVD and prevent further disc degeneration [21,22,23,24].

To explore the role of the immune system in the retention of the injected cells we used two rat models, immunodeficient Nude rats and immunocompetent Sprague Dawley (SD) rats. To quantify cell retention, cells were labeled with DiI lipophilic dye or transduced with green fluorescent protein (GFP) prior to the injection. Cell survival and systemic inflammation was assessed in both rat types by means of flow cytometry, enzyme-linked immunosorbent assay (ELISA), histological and immunohistochemical analyses. We hypothesized that both the knee joint and the IVD are immune privileged sites and that cell retention is not affected by presence or absence of an immune system in the injected animals.

## 2. Materials and Methods

### 2.1. Study Design

For the ACL tear model (Figure 1A), iPSCs were differentiated to iMSCs as previously reported [25]. iMSC were labeled with lipophilic DiI dye (iMSC^DiI+^) to allow for cellular identification following injection. To test immune system reaction to xenotransplantation of cells after knee injury, Nude (*n* = 5) and SD (*n* = 10) rats underwent a rupture of their right ACL using a standardized non-invasive knee injury two weeks prior to the injection of 1 × 10^6^ iMSC^DiI+^ into both knee joints of each rat. To assess systemic inflammation markers, whole blood was repetitively collected on the day of injection (day 0), as well as on days 3, 7, 10 and 14. Two weeks after the cells were injected, the animals were sacrificed and knee tissues were processed for flow cytometry, histological and immunohistochemical analyses.

For the IVD model (Figure 1B), IVDs were collected from human donors to generate GFP-overexpressing NPCs (NPC^GFP+^) according to standard protocols as previously reported [26,27,28]. Shortly before their injection, NPC^GFP+^ were further labeled with lipophilic DiO dye to allow cellular tracking following transplantation. X-ray guidance was used to inject 5 × 10^5^ NPC^GFP+/DiO+^ into two lumbar IVDs (L3/4 and L4/5) of Nude (*n* = 4) and SD (*n* = 5) rats. To assess systemic inflammation markers, whole blood was repetitively collected on the day of injection (day 0) as well as on days 2 and 7. After one week, the animals were sacrificed and IVD tissues processed for flow cytometry, histological and immunohistochemical analyses.

### 2.2. Non-Invasive Anterior Cruciate Ligament (ACL) Rupture

To test immune system reaction to xenotransplantation of cell in context of a knee injury (PTOA model), all animals underwent a rupture of their right ALC by means of a non-invasive tear (Figure 2A–D) as previously described [29]. Briefly, rats were anesthetized using 4% isoflurane in oxygen and placed inside a custom-built jig, which was secured to the frame of the hydraulic mechanical testing system (370.02 Bionix, MTS Systems Corp., Eden Prairie, MN, USA). Force and displacement of the actuator were continually recorded on the MTS data acquisition system at 128 Hz. ACL rupture was determined by a 20% drop in tibial compression. Ultimate load at maximum tibial displacement at failure was determined from load-displacement curves generated from the MTS data to confirm consistent ACL rupture (Figure 2C,D). These measurements served the purpose of confirming comparable rupture conditions between both rat type groups. After a successful rupture of the ACL was confirmed using the Lachman’s test, which assesses the degree of excessive anterior tibial translation, the knees further underwent X-ray imaging to exclude potential bone fractures. Subsequently, the animals were returned to their cages and monitored until fully recovered from anesthesia.

### 2.3. Generation of iMSCs from iPSC

For the PTOA model, human iPSCs were obtained from the Cedars-Sinai iPSC core facility and differentiated towards their mesenchymal lineage (iMSC, Figure 2E) using our previously published protocol [30]. First, control human dermal fibroblasts were derived from healthy donors at the Cedars Sinai Medical Center and a virus-free iPSC line subsequently produced using a Human Fibroblast Nucleofection Kit (Lonza, Portsmouth, NH, USA). Briefly, 0.8 × 10^6^ fibroblasts per nucleofection were harvested and centrifuged at 200× *g* for 5 min. After the supernatant was removed and the cell pellet resuspended in Nucleofector Solution (VPD-1001; Lonza, Portsmouth, NH, USA), the episomal plasmid expression of six factors (OCT4, SOX2, KLF4, L-MYC, LIM28, and p53 shRNA) was added. Compared to the approach of a viral transduction, this method bears the advantage of gene expression in a transient manner rather than being integrated. After the cell/DNA suspension was transferred back into Nucleofection solution and maintained under 5% O_2_ for 48 h, the culture medium was gradually changed to human iPSC (hiPSC) medium, which included sodium butyrate, glycogen synthase kinase 3β inhibitor of the Wnt/β-catenin signaling pathway, a mitogen-activated protein kinase pathway inhibitor and a selective inhibitor of TGF-β type I receptor ALK5 kinase, type I activin/nodal receptor ALK4, as well as type I nodal ALK7.

### 2.4. Isolation of Human NPCs

For the IVD model, discs of human donors were collected from the Cedars-Sinai Medical Center biobank. Subsequently, the nucleus pulposus (NP) tissue was extracted, mechanically minced into small pieces (~1 mm) and washed 4 times with phosphate-buffered saline (PBS). Next, the NP tissue was digested for one hour at 37 °C by means of digestion medium (DMEM/F12 + 1% Antibiotic/Antimycotic) including 2 mg/mL of Pronase Protease (Sigma Aldrich, St. Louis, MO, USA, Cat # 53702). Afterwards, the digested tissue was centrifuged for 10 min at 1200 RPM, and the supernatant was discarded. The remaining pellet was then further digested overnight at 37 °C, using digestion media including 110 mg/mL collagenase Type 1S (Sigma Aldrich, Cat #C1639). The resulting suspension was filtered through a 70-µm filter and centrifuged once again for 10 min at 1200 RPM at room temperature. Once the supernatant was discarded, the remaining cells were counted and plated in a 100 mm petri dish at a concentration of 1 × 10^6^ cells per plate (Figure 2F).

### 2.5. Genetic Engineering of Human NPC to Overexpress GFP

To generate GFP-overexpressing cells, NPC were engineered using a lentiviral vector. Briefly, HEK293T/17 cells (ATCC, Manassas, VA, USA) were seeded at a density of 60,000 cells/cm^2^ in Eagle’s Minimum Essential Medium (EMEM; #30-2003, ATCC) containing 10% fetal bovine serum (FBS; #100-106, Gemini Bio, West Sacramento, CA, USA) and 1% antibiotic-antimycotic solution (AAS; #15240096, ThermoFisher, Waltham, MA, USA). After 24 h, the lentivirus was produced by transfecting the HEK293T/17 cells with the mGFP viral vector and two packaging plasmids (pCMV-dR8.2, pCMV-VSV-G; all three plasmids were gifted by the Simon Knott laboratory at Cedars-Sinai Medical Center). The transfections were conducted using the BioT method with a 1.5:1 ratio of BioT (μL) to DNA (μg). The virus-containing medium was harvested at 48 and 72 h after transfection, centrifuged, filtered and frozen into aliquots. The aliquots were thawed and immediately used to transduce the NPC. Transduction titers for GFP were determined using flow cytometry.

### 2.6. Fluorescent Labeling of iMSC and NPC^GFP+^

To assure successful tracking of cells upon their injection in vivo, highly lipophilic DiI (iMSC) and DiO (NPC^GFP+^) cell-labeling solutions were used for the cell staining, according to manufacturers’ instructions (Figure 2G,H). Briefly, the cells were washed with PBS, detached from their plates, collected, centrifuged at 1400 RMP for 6 min, and the resulting cell pellet was resuspended in a final concentration of 1 × 10^6^ cells/mL. For each mL of cell-suspension prepared, 5 µL of either DiI (PTOA model) or DiO (IVD model) were added, and the cells incubated for 5 min under the absence of light at 37 °C. Subsequently, the cells were further stored under the absence of light at 4 °C for 15 min. After a total of 20 min, the stained cell suspensions were centrifuged for 6 min at 1400 RPM at 24 °C. Lastly, the supernatant was removed, and the cells were resuspended in a final volume of PBS. The final cell suspension was stored on ice until being injected into the corresponding anatomical site of the animal model.

### 2.7. X-ray-Guided Injection of iMSC^DiI+^ and NPC^GFP+/DiO+^

To ensure an accurate and standardized transfer of cells into the corresponding anatomical target structures, the injection process for both experimental models 1 and 2 was performed under sterile conditions, inhaled general anesthesia (4% isoflurane in oxygen) and X-ray C-arm guidance (Fluoroscan^®^ InSightTM 2 Mini C-Arm, Hologic, Marlborough, MA, USA). For both experiments, Appropriate Institutional Animal Care and Use Committee (IACUC #010218 (PTOA) and #008066 (IVD)) approval was obtained before conducting the surgical operations.

For knee injection of iMSC^DiI+^ (PTOA model), the anesthetized animals were placed on a heated surface in supine position after their knees were shaved. The leg was then carefully stretched, and the foot fixed to the underlying surface with tape to maintain a stable position. Subsequently, a 27 G needle attached to a 100 µL Hamilton syringe was used to inject a 25 µL cell solution consisting of 20 µL PBS, 5 µL Omnipaque contrast agent, and 1 × 10^6^ iMSC^DiI+^ (Figure 2I). After the needles were carefully removed, the rats were returned into their cages with adequate access to food and water.

To inject NPC^GFP+/DiO+^ (IVD model) into L3/4 and L4/5 IVD, the lumbar spine was first exposed, and the lumbar discs identified using both pelvic rim and aortic bifurcation as anatomical landmarks. Shortly before the injection of cells, the precise position of the needle inside the discs was confirmed with X-ray images taken from both a lateral and supine position (Figure 2J,K). Once the needle tip was placed in the center of the IVD, 8 µL PBS including 5 × 10^5^ NPC were carefully injected. After both peritoneum and skin were closed, rats were each placed in a solitary recovery cage with adequate access to food and water.

The animals of both the PTOA and IVD model received antibiotics (Baytril; 5 mg/kg bodyweight) and narcotics (Buprenorphine; 0.05 mg/kg bodyweight) subcutaneously for 2 days postoperatively.

### 2.8. Blood Collection and Serum Preparation

In order to test the animals for systemic inflammation markers, 0.8 mL of whole blood was repetitively collected per animal on days 0 (day of cellular injection), 3, 7, 10 and 14 (PTOA model) as well as on days 0 (day of cellular injection), 2 and 7 (IVD model), using a 0.8 mL serum separator tube (Greiner Bio One MiniCollect, Monroe, NC, USA). Collected samples were subsequently centrifuged at 2000× *g* for 10 min. Lastly, the supernatant (serum) was removed and stored at −20 °C until being used.

### 2.9. Tissue Extraction and Digestion

In order to analyze the cellular composition of the synovial joints and IVDs following the injection of iMSC^DiI+^ and NPC^GFP+/DiO+^, respectively, knee and lumbar disc tissues were extracted and digested, and single cell solutions were isolated and processed for flow cytometry.

For the PTOA model, both knee synovium knee tissues per rat were isolated, placed into a 1.5 mL tube (Eppendorf, Framingham, MA, USA) and physically minced with fine scissors for 5 min. After achieving a homogeneous structure, 1 mL of DMEM medium, including 5 µL of collagenase type II (Millipore Sigma, Burlington, MA, USA), was used to enzymatically digest the tissue in a rocker for 1.5 h at 37 °C. Subsequently, the samples were vigorously vortexed for 1.5 min and centrifuged at 1200 RPM for 10 min. After the supernatant was aspirated, the remaining tissue was re-suspended in 1 mL PBS, filtered through a 35 µm mesh, counted and processed for flow cytometric analyses.

For the IVD model, lumbar L3/4 and L4/5 IVD synovial tissues were extracted and morselized with fine scissors. Subsequently, 5 mL of NPC processing media (DMEM/F12 + 10% FBS + 1% ascorbic acid) was added, including 65 µL Pronase (Fisher Scientific, Waltham, MA, USA), and enzymatically digested for 1 h at 37 °C on a rocker. Thereafter, the tissue solution was centrifuged for 10 min at 30× *g*, and the supernatant was aspirated. In a second step of enzymatical digestion, the remaining tissue was re-suspended in 1 mL of NPC processing medium (DMEM/F12 + 10% FBS + 1% ascorbic acid), including 125 µL of collagenase type I-S (110 mg/mL; Millipore Sigma, Burlington, MA, USA) for 4 h at 37 °C on a rocker. Following the digestion, the cell suspension was filtered through a 35 µm mesh and processed for flow cytometric analyses.

### 2.10. Flow Cytometry

To determine possible immune response to the injury, we assessed levels of host CD8^+^ cells and fluorescently labeled (GFP and DiO or DiI) transplanted cells recovered from the rats in both models. After the cells were isolated, they were washed with FACS buffer containing 2% bovine serum albumin (BSA, A4503, Sigma, St. Louis, MO, USA) and 0.1% sodium azide (S2002, Sigma) in PBS. The cells were either, unstained or labeled with CD8^+^ anti-rat antibody (APC, # 17-0084-82, Invitrogen, Carlsbad, CA, USA) for 15 min protected from light at 4 °C. After the incubation, cells were washed again and resuspended with FACS buffer. A BD LSR Fortessa analyzer (BD Biosciences) was used to acquire the data of CD8^+^ signal and of the DiI signal that was previously introduced to injected cells. Afterwards, the data was analyzed using FlowJo software (FlowJo LLC, Ashland, OR, USA).

### 2.11. ELISA

Quantification of IL-6 cytokine levels in the serum a rat IL-6 Rat ELISA Kit (BMS625; Life Technologies, Carlsbad, CA, USA). To determine the presence of IgM in the serum samples, a Rat IgM ELISA kit was used (MBS2510638; MyBioSource, San Diego, CA, USA). For IL-6 and IgM assays, samples were allowed to thaw at room temperature and were not diluted in case of IL-6 and diluted 1:10,000 in case of IgM (according to the manufacturer’s instructions) when the assays were performed. Standard curves were performed in accordance with the standard values indicated by the manufacturer of the ELISA kits. All data sets were exported to Excel, and concentrations were calculated using interpolation in Prism 8 software (GraphPad, La Jolla, CA, USA).

### 2.12. Histology and Immunofluorescence

Two weeks (PTOA model) or one week (IVD model) post-injections, knee joints (*n* = 5) and IVD tissues (*n* = 3) were harvested for histological analysis and immunofluorescence imaging. Shortly upon sample retrieval, knee and IVD tissues were placed into 4% Formaldehyde solution and fixed for 48 h at room temperature. Subsequently, the samples were washed with tap water and decalcified using ethylenediaminetetraacetic acid (EDTA), which was changed once a week until the samples appeared soft enough to be easily cut. Once the appropriate stiffness was achieved, knee and IVD tissues were cut in half, embedded in paraffin and cut into 5 µm-thick sections. A Hematoxylin and Eosin (H&E) staining was performed to illustrate morphological features of the tissue types after cellular injections.

Furthermore, immunofluorescence imaging was used to visualize DiI- and DiO-labeled cells within deparaffinized histological tissue slides of knee and IVD target tissues, using a Revolve Microscope (model RVL-100-B2, ECHO Laboratories, San Diego, CA, USA) coupled with the Echo Pro Software. Images were taken using 20× magnification and an exposure time of 600 ms (Cy3 filter, PTOA model) and 420 ms (FITS filter, PTOA model).

### 2.13. Statistical Analysis

All statistical analyses were performed via Prism 8 (GraphPad, La Jolla, CA, USA) with *p* < 0.05 noted as statistically significant. Assessed outcome measurements were biomechanical properties, flow cytometric cell count and ELISA protein expression. Data points over two standard deviations were excluded from analysis. For biomechanical properties, a non-parametric *t*-test was performed for each of the parameters separately. For flow cytometric cell counts and ELISA protein expression, 2-way analysis of variance (ANOVA) was performed for each dependent measure separately, using mean values with grouping of experimental groups. In the figures, average (+/− standard deviation) values are shown. Data values that were greater or lower than two times the standard deviation were defined as outliers and excluded from the final data set.

## 3. Results

To avoid typical xenotransplantation-induced immunological or physiological barriers, small animal models with a genetically modified background are widely used [8,31,32,33,34]. Particularly immunocompromised, t-cell-deficient Nude rats are frequently utilized in comparison to immunocompetent SD rats when it comes to analyzing the behavior of body-foreign cell types in rodents [35,36,37]. Commonly used cell dyes (DiI/DiO) were further used to enable the tracing of injected cells in vivo (Figure 1) [38,39]. Particularly for the investigation of PTOA-related cellular mechanisms, non-invasive knee injury was used to rupture the right ACL of the animals, which, in turn, destabilizes the knee and allows the continuous development of PTOA. No significant differences were detected in biomechanical parameters, such as ultimate load (N), maximum displacement (mm), stiffness (N/mm) and toughness (N × mm) of both Nude and SD rats, conforming the induction of a comparable injury (Figure 2C–F).

### 3.1. Recovery of Injected Cells

The joint (PTOA model) and the IVD (IVD model) tissues were digested as previously described, the single cell suspensions were used to analyze the cellular composition within both target organs following their injection with iMSC^DiI+^ or NPC^GFP+/DiO+^, respectively.

For the PTOA model, a comparable number of cells could be isolated from uninjured and injured knees as well as from Nude and SD rats (Figure 3A). Using this recovered cell population, flow cytometric assessment further revealed a comparable percentage of CD8^+^ T-cells within the joints of both uninjured and injured knees. Of interest, the percentage of CD8^+^ cells were also comparable between the two rat types (Figure 3B). Lastly, the percentage of DiI^+^ cells within the recovered cell population was assessed to determine the survival rate of injected cells 2 weeks post-injection. In line with the CD8 assessment, no significant differences in the percentage of DiI^+^ cells were detected between the uninjured and injured knees of the rat types (Figure 3C).

### 3.2. Immune Response to Injected Cells

To determine the systemic inflammation of both Nude and SD rats following xenotransplantation over time, repetitively collected whole blood was used to generate blood serum for ELISA analyses.

For the PTOA model, ELISA has revealed a comparably low level of IL-6 inflammation marker between Nude and SD rats on day 0 (day of cellular injection) as well as on days 7 and 10 after injection. On day 14 following the xenotransplantation of iMSC^DiI+^, Nude rats show a significantly elevated level of IL-6 when compared to SD rats (Figure 4A). Furthermore, ELISA was used to quantify the level of IgM as a main regulator for inflammatory responses. Although Nude rats revealed a significantly elevated systemic level of IgM on the day of cellular injection (day 0), no differences were detected throughout the following time course in days 7, 10 and 14 (Figure 4B).

For the IVD model, Nude and SD rats revealed a comparably low level of IL-6 on days 1 and 7 following the xenotransplantation of NPC^GFP+/DiO+^ (Figure 4C). Additionally, there were no significant differences in systemic IgM levels of both rat types, indicating a comparably low immune response to the injected cells (Figure 4D). In line with these findings, other studies confirmed IVD tissue as very tolerant towards xenotransplanted cells. For example, Henrikkson et al. (2009) have shown that no inflammatory cell reactions were observed after the injection of human MSCs into lumbar discs of mini-pigs [40]. Furthermore, Iwashina et al. (2006) demonstrated that no cell-induced inflammation was detected at the site of injury after the xenotransplantation of human NPCs into degenerated rabbit discs [41].

### 3.3. Histological Detection of Dye-Labeled Cells

H&E staining was used to confirm the intact morphology of the knee (Figure 5A,B) and IVD (Figure 5C,D) tissues following the xenotransplantation of 1 × 10^6^ iMSC^DiI+^ or 5 × 10^5^ NPC^GFP+/DiO+^, respectively.

Furthermore, to confirm the functionality of the model as well as the stability and presence of the cell staining dyes over time, immunofluorescent imaging illustrating retention of iMSC^DiI+^ (Figure 5E–G) and NPC^GFP+/DiO+^ (Figure 5H–J) at the end of the experimental runs.

## 4. Discussion

Small animal models are commonly used to systemically study cellular retention and inflammation following xenotransplantation [42]. When it comes to the effect of host tissue types towards xenotransplanted cells, inconsistent literature and conflicting observations [8,9,10,11,12] clearly indicate the need for further analyses. Accordingly, this present study analyzed the retention and immune response effects of human iMSC^DiI+^ and NPC^GFP+/DiI+^ after their xenotransplantation into particularly immunoprivileged sites, knee synovial joints and IVDs of immunocompromised Nude and immunocompetent SD rats.

Flow cytometric assessment of tissue samples extracted from both study sections demonstrated comparable percentages of cell recovery of ~10% (PTOA model, Figure 3A) and ~30–50% (IVD model, Figure 3D) in the injected tissues of Nude and SD rats. Further flow cytometric investigation revealed that ~30% (PTOA model, Figure 3B) and ~15% (IVD model, Figure 3E) of those recovered cells were CD8 positive, a commonly used marker for cytotoxic t-lymphocytes that mediates an adequate interaction between cells of the immune system [43,44,45,46]. These numbers are in line with results of other research groups that tested the immune response of xenotransplanted cells [47,48,49,50]. Especially for xenotransplantation approaches, these regulatory T-cells are described to be of highest relevance to transplantation success [51]. Of interest, slight but statistically non-significant differences in IL6 and IgM levels can be seen between the two lumbar discs (Figure 4C,D) of the IVD model. Since both lumbar discs were treated exactly identical, such observed variations may result from high methodical variability rather than from different immunological responses.

Although such an immune response would generally be expected to be higher in immunocompetent SD rats when compared to immunodeficient Nude rats, both rat types revealed a comparable immune response, indicating that both knee and IVD tissues exert their immunoprivileged capabilities following the xenotransplantation of cells. Lastly, ~25% of all recovered cells in the PTOA model and ~4% of all recovered cells in the IVD model were DiI^+^ and GFP^+^, respectively (Figure 3C,F). This indicates the successful survival of injected cells for the duration of the experiment.

To further assess the inflammation level of both rat types following the xenotransplantation of iMSC^DiI+^ and NPC^GFP+/DiI+^, blood serum was used to quantify the systemic IL-6 and IgM levels using ELISA. Particularly IL-6 is known to be able to promote dysregulation of coagulation and systemic inflammation after xenotransplantation [32]. Whereas comparable levels of IL-6 could be detected within Nude and SD rats on days 7 and 10 following the xenotransplantation of iMSC^DiI+^ into knee joints, a significantly elevated level in the serum of Nude rats was detected on day 14 when compared to SD rats (Figure 4A). Although a higher immune response would be expected in immunocompetent SD rats, IL-6 elevation can be seen as a typical feature of immunocompromised Nude rats [52]. In fact, immunocompromised rats tend to compensate their T-cell deficiency by means of a macrophage-induced overproduction of inflammatory cytokines, such as IL-6, which is crucially needed to promote T-cell proliferation and differentiation. Additional IgM detection in systemic blood serum revealed a comparable amount of ~1 µg/mL IgM throughout the entire 2-weeks experiment (Figure 4B). In combination with the overall comparable levels of systemic IL-6 and IgM, these findings clearly indicate that the xenotransplantation of iMSC^DiI+^ did not result in an elevated inflammatory response within SD rats when compared to Nude rats.

In line with our findings of the PTOA model, ELISA of systemic IL-6 and IgM levels in rats injected with NPC^GFP+/DiO+^ into the IVD also shows comparable results between both rat types, indicating that the xenotransplantation of human primary cells into immune privileged lumbar discs does not lead to an elevated immune response and inflammation.

This study is not without limitations. Only one cellular load as well as one relatively short time frame has been tested in each experimental section. Although the goal of this study was to observe the animal’s rejection of xenotransplanted cells as well as their inflammatory behavior during the particularly acute phase of inflammation, which occurs within the first hours and days after xenotransplantation or injury [53,54,55,56], longer time periods must be analyzed to gain a broader overview. Further studies should not only aim for a longer observation period but should also analyze multiple cell numbers to determine the highest cellular load per anatomical structure that may be tolerable without invoking severe immune response.

## 5. Conclusions

Conclusively, the results of our study show that the injection of iMSC^DiI+^ and NPC^GFP+/DiI+^ into immunoprivileged knee and IVD anatomical sites did not lead to an elevated inflammatory response in SD rats when compared to Nude rats. Hence, SD rats represent suitable animals for further xenotransplantation studies, targeting immunoprivileged sites.

## Figures and Tables

**Figure 1 bioengineering-10-01049-f001:**
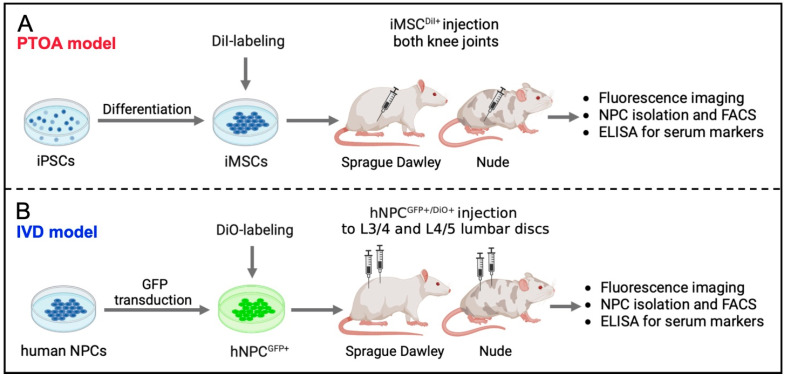
Experimental outline. (**A**): Schematic overview of the PTOA model. Purchased iPSCs are differentiated into their mesenchymal lineage (iMSCs) and labeled with lipophilic DiI dye. Subsequently, 1 × 10^6^ iMSC^DiI+^ were injected into both knees of Nude (*n* = 7) and SD (*n* = 7) rats that underwent a non-invasive rupture of their right ACL 2 weeks prior. After 2 weeks, animals were anesthetized, and knee synovium tissues were processed for flow cytometric analyses and ELISA as well as histology and immunofluorescent imaging. (**B**): Schematic overview of the IVD model. NPCs of human donors were cultured, transduced with GFP and labeled with lipophilic DiO dye. Subsequently, 5 × 10^5^ NPC^GFP+/DiO+^ were injected into each L3/4 and L4/5 lumbar discs of Nude (*n* = 4) and SD (*n* = 5) rats. After 1 week, animals were anesthetized, and lumbar disc synovium tissues processed for flow cytometric analyses and ELISA as well as histology and immunofluorescent imaging.

**Figure 2 bioengineering-10-01049-f002:**
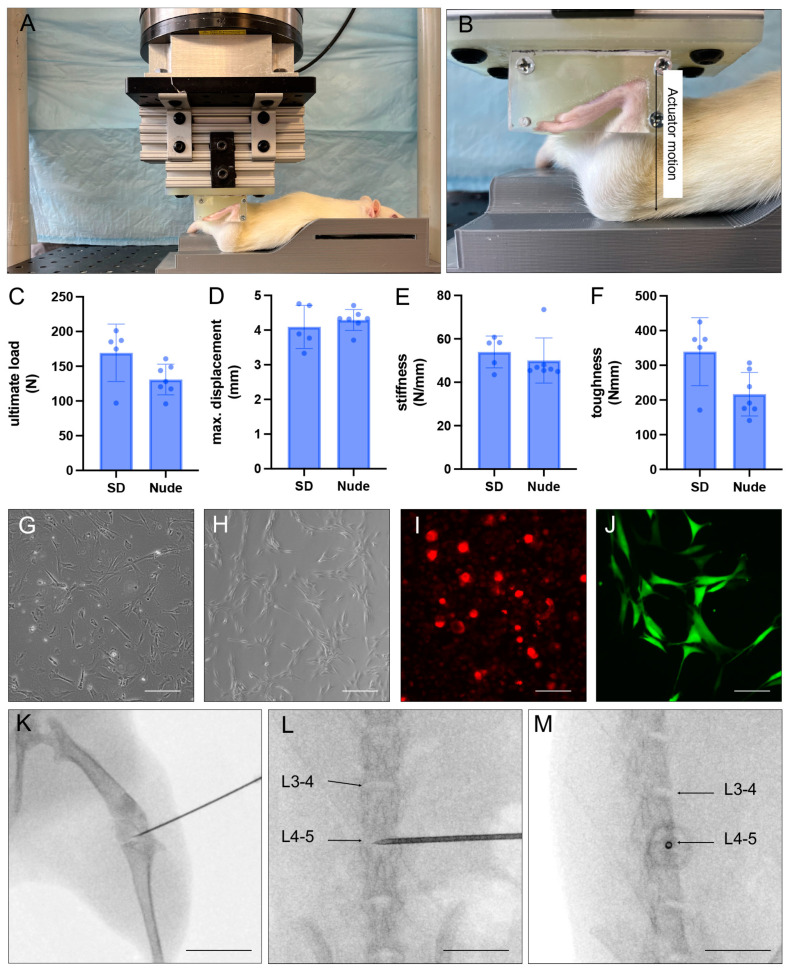
Materials and methods. (**A**,**B**): Representative images of a SD rat during non-invasive ACL rupture. (**C**–**F**): Biomechanical parameters (ultimate load (**C**), maximum displacement (**D**), stiffness (**E**), and toughness (**F**)) during non-invasive ACL rupture of Nude and SD rats for the PTOA model. Means  ±  SD. (**G**,**H**): Representative images of iMSCs (**G**) and NPCs (**H**) in culture prior to the labeling with lipophilic dyes. Scale bars = 100 µm. (**I**,**J**): Representative images of iMSC^DiI+^ (**I**) and NPC^GFP+/DiO+^ (**J**) in culture shortly before their xenotransplantation. Scale bars = 100 µm. (**K**–**M**): Representative images of X-ray C-Arm-guided injection of cells into the knee joints (PTOA model; (**K**)) and lumbar discs (IVD model; (**L**,**M**)) of Nude and SD rats. Scale bars: K = 13 mm, L + M = 8 mm.

**Figure 3 bioengineering-10-01049-f003:**
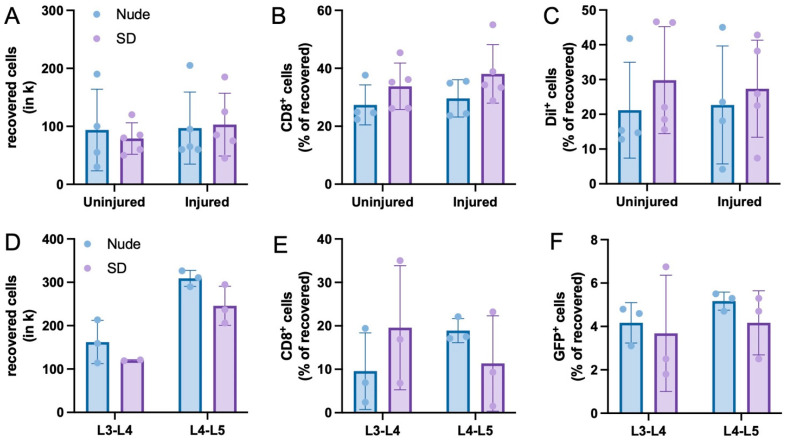
Flow cytometry. (**A**–**C**): Total number of recovered cells (**A**) as well as the percentage of CD8^+^ (**B**) and DiI^+^ (**C**) cells in extracted tissue of the PTOA model. (**D**–**F**): Total number of recovered cells (**D**) as well as the percentage of CD8^+^ (**E**) and DiI^+^ (**F**) cells in extracted tissue of the PTOA model. Means  ±  SD. For the IVD model, a comparable number of cells could be isolated from L3-L4 and L4-L5 lumbar discs with no significant differences between the rat types (**D**). Using this recovered cell population, flow cytometric assessment further revealed a comparable percentage of CD8^+^ T-cells within both L3-L4 and L4-L5 lumbar discs after injection (**E**). Lastly, the percentage of GFP^+^ cells within the recovered cell population was assessed to determine the survival rate of injected NPC^GFP+/DiO+^ after 1 week. The percentage of GFP^+^ cells within L3-L4 and L4-L5 lumbar discs was comparable between both Nude and SD rats (**F**).

**Figure 4 bioengineering-10-01049-f004:**
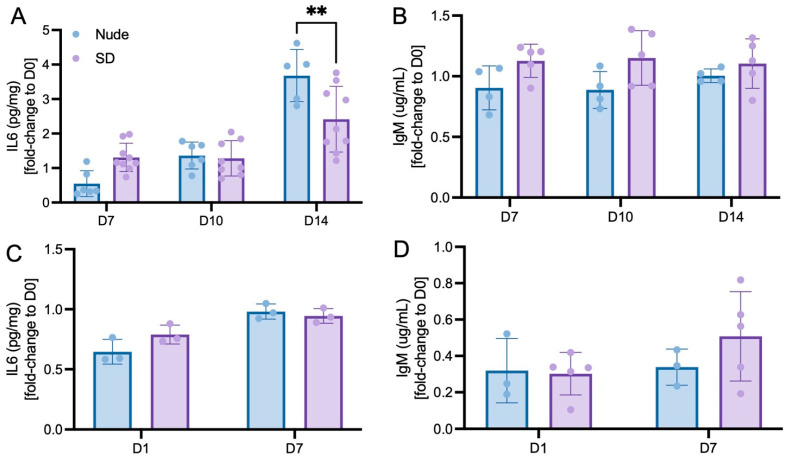
(ELISA). (**A**,**B**): Systemic IL-6 (**A**) and IgM (**B**) levels (pg/mg) of Nude and SD rats throughout the 14-days experimental run of the PTOA model. Means  ±  SD. ** *p*  <  0.001 vs. Nude. (**C**,**D**): Systemic IL-6 (**C**) and IgM (**D**) levels (pg/mg) of Nude and SD rats throughout the one-week experimental run of the IVD model. Means  ±  SD.

**Figure 5 bioengineering-10-01049-f005:**
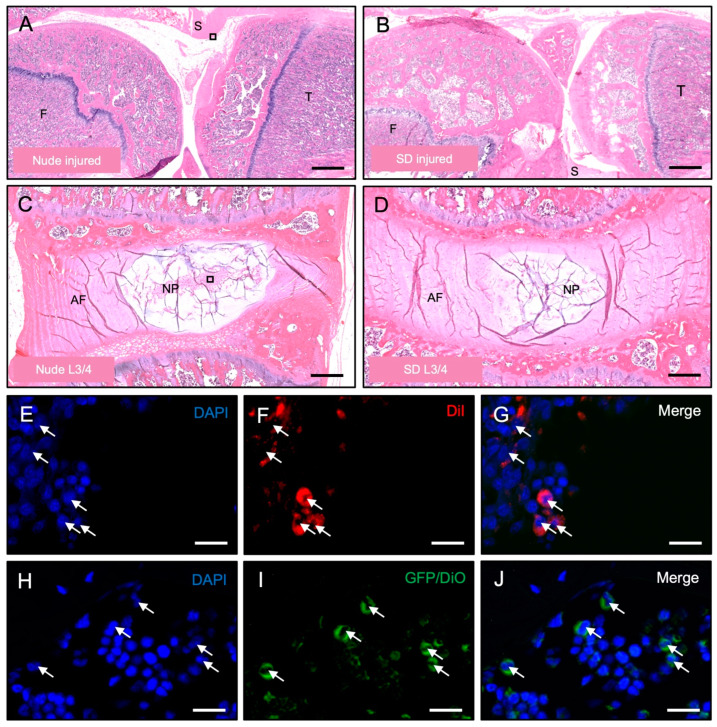
Histology and immunofluorescence. (**A**–**D**): Representative images of knee (**A**,**B**) and lumbar disc (**C**,**D**) tissue of Nude (**A**,**C**) and SD (**B**,**D**) rats at the end of the experimental run (F = Femur, T = Tibia, AF = Annulus Fibrosus, NP = Nucleus Pulposus). Scale bars: A + B = 1 mm, C + D = 300 µm. (**E**–**G**): Representative images of DiI^+^ cells (arrows) inside knee tissue at the end of the 2-week experimental run of the PTOA model. Scale bars = 15 µm. (**H**–**J**): Representative images of DiO^+^ cells (arrows) inside the tissue of lumbar discs at the end of the 1-week experimental run of the IVD model. Scale bars = 15 µm.

## Data Availability

Data is contained within this article.

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
