# Peer review of "Retention of Human iPSC-Derived or Primary Cells Following Xenotransplantation into Rat Immune-Privileged Sites"

_bioengineering, 2023, doi:10.3390/bioengineering10091049_

Round 1

Reviewer 1 Report

This study utilizes the PTOA and IVD models to investigate the transplantation of human cells into rats. There are some issues that need to be clarified.     

1.    The title “2.3. Generation of MSCs from iPSC” should be should be corrected as “2.3. Generation of iMSCs from iPSC”

2.    Page 8, Lines 299-302

“No significant differences were detected in 299 biomechanical parameters, such as ultimate load (N), maximum displacement (mm), stiff- 300 ness (N/mm), and toughness (N x mm) of both Nude and SD rats, conforming the induction of a comparable injury (Figure 2C-F).” There is no data in the Figure 2. C+D

3.    In Figure 3A, the PTOA model shows that there are five data points for cell recovery from Nude mice, while the others have only four. What happened to the missing sample in the PTOA Nude data, and is the same observation present in Figure 3D for the IVD model in SD rats at L3-L4?

4.    CD8-positive cells can be considered as markers of immune expression. In Figure 3E, there appears to be no difference between Nude mice and SD rats; however, some SD rats still exhibit higher levels of CD8-positive cells in L3-L4 and L4-L5, respectively. How can this condition be explained?

5.    In Figures 4A and 4B, the number of Nude mice is inconsistent. In Figure 4A, regarding the SD rat group, the numbers are inconsistent for D7 (n=9), D10 (n=8), and D14 (n=9). Additionally, in Figure 4B, there are only 5 SD rats shown in the figure. Similarly, there is also inconsistency in the number of SD rats depicted in Figures 4C and 4D. Although Nude rats reveled a significantly elevated systemic level of IgM on the day of cellular injection (day 0), How to explain that higher IgM level in the Nude rats than the SD rat at day 0?

6.    The legends of Figures 4A and 4B indicate '*p<0.05'. Please mark the corresponding significance on the figures.

Author Response

The authors would like to thank reviewer 1 for his contribution and helpful questions, which made this publications better. A detailed response letter can be found attached.

Reviewer 2 Report

In this manuscript the authors show that human cells (notably supposedly  hypoimmunogenic mesenchymal stem cells) can be transplanted to two immunoprivileged sites in a rat model, regardless of the rat being immunocompromised or not, just suggesting possible therapeutic applications with non-matched cells if they are introduced to immunoprivileged site. Overall this is very interesting, and the manuscript is well organized and straightforward. I have only a few comments.

- I am missing a representative positive control for these experiments, showing that cell injection into non-immunoprivileged sites causes the appropriate immune reaction it is supposed to, and that this reaction is accurately detected/monitored using the methodology applied here.  Likely the authors already have this information.

-The images shown in Figure 5 E-J could be quantified to see in human cells survival in each of the different models matches the flow cytometru data (Fig 3).

-The recovery of labelled cells seems very low in one of the experiments, compared to the other (Fig 3C vs Fig 3F, specifically).  Do the authors have an explanation for this difference? And can this be an issue in terms of interpreting the findings? This should be discussed.

Author Response

The authors would like to thank reviewer 2 for the questions. A detailed response is provided attached.

Round 2

Reviewer 1 Report

Please see the attachment. Thank you
